# Machine Learning for Postoperative Continuous Recovery Scores of Oncology Patients in Perioperative Care with Data from Wearables

**DOI:** 10.3390/s23094455

**Published:** 2023-05-02

**Authors:** Meike A. C. van den Eijnden, Jonna A. van der Stam, R. Arthur Bouwman, Eveline H. J. Mestrom, Wim F. J. Verhaegh, Natal A. W. van Riel, Lieke G. E. Cox

**Affiliations:** 1Philips Research, 5656 AE Eindhoven, The Netherlands; 2Department Biomedical Engineering, Eindhoven University of Technology, 5612 AE Eindhoven, The Netherlands; 3Department of Clinical Chemistry, Catharina Hospital, 5602 ZA Eindhoven, The Netherlands; 4Department of Anaesthesiology, Catharina Hospital, 5602 ZA Eindhoven, The Netherlands; 5Department of Electrical Engineering, Eindhoven University of Technology, 5612 AE Eindhoven, The Netherlands

**Keywords:** post-operative recovery, wearable sensors, machine learning, hospital discharge, oncology, physical activity, vital signs, monitoring, clinical prediction

## Abstract

Assessing post-operative recovery is a significant component of perioperative care, since this assessment might facilitate detecting complications and determining an appropriate discharge date. However, recovery is difficult to assess and challenging to predict, as no universally accepted definition exists. Current solutions often contain a high level of subjectivity, measure recovery only at one moment in time, and only investigate recovery until the discharge moment. For these reasons, this research aims to create a model that predicts continuous recovery scores in perioperative care in the hospital and at home for objective decision making. This regression model utilized vital signs and activity metrics measured using wearable sensors and the XGBoost algorithm for training. The proposed model described continuous recovery profiles, obtained a high predictive performance, and provided outcomes that are interpretable due to the low number of features in the final model. Moreover, activity features, the circadian rhythm of the heart, and heart rate recovery showed the highest feature importance in the recovery model. Patients could be identified with fast and slow recovery trajectories by comparing patient-specific predicted profiles to the average fast- and slow-recovering populations. This identification may facilitate determining appropriate discharge dates, detecting complications, preventing readmission, and planning physical therapy. Hence, the model can provide an automatic and objective decision support tool.

## 1. Introduction

After surgery, patients recover along different paths, which may be smooth or include complications, resulting in a specific hospital length of stay (LOS) for each patient. Assessment of post-operative recovery is important in the clinic as it can reduce unnecessary time spent in the hospital as well as the number of preventable readmissions by finding appropriate discharge dates [1]. A prolonged stay in the hospital is costly and can be stressful for patients and their families [2], whereas sending patients home too early imposes the risk of patients deteriorating at home [3]. Furthermore, accurate recovery assessment could assist in identifying patients who require additional care, and it might provide early warning triggers for complications [4]. As a result, it may lead to improved patient care planning, surgery planning, resource utilization, and an increase in bed capacity [1,5].

Several scales exist for reporting functional recovery outcomes and disability after hospitalization. However, these scales are not standardized, are disease-specific, contain unclear outcome categories, and are unreliable and unprecise [6]. Recovery-based decisions, such as discharge readiness or patient stability, are thus often complex and highly subjective in nature [3]. To support identifying patients with complex or delayed recovery, early warning scores (EWSs) can be utilized [7]. EWSs utilize multiple parameters, including vital signs (heart rate, respiration rate, systolic blood pressure, and temperature) and non-vital signs (level of consciousness). These parameters are measured with spot checks during nursing rounds approximately three times a day. A downside of this approach is the discontinuity, as all changes in parameters between these spot checks remain unrecorded. Furthermore, this system is nonautomatic, making it time consuming and labor intensive for the nurses to collect and register the parameters.

Despite the limitations of current methods, investigations to improve the assessment of post-operative recovery have been rather unaddressed in the literature. Previous studies have often investigated predicting events such as discharge or complications, instead of extensive recovery trajectories. For example, for oncology patients, research has focused on predicting LOS, complications, and mortality [8,9]. By predicting these in-hospital events, previous research has often focused on short-term recovery, defined as recovery from surgery until discharge [10]. However, little research has investigated recovery after hospital discharge [10], which is important to investigate as well, as deterioration at home occurs.

To overcome the limitations of the current methods and research, wearable sensors can be utilized, since these sensors measure continuously with limited personnel burden [11,12]. Due to the objectivity and the validity of the devices, the sensors are likely to be unbiased and reliable [6]. Moreover, the sensors monitor regardless of the patient’s location, so they can also be suitable for measuring both in the hospital and at home [6]. Wearable sensors can be used to measure, for example, vital signs and activity metrics. Vital signs are likely to be significant indicators of post-operative recovery, as they are accepted measures in the hospital and they can detect delayed recovery [13]. Activity metrics can also be considered a marker of functional post-operative recovery [11], and they have been shown to be related to LOS, functional decline, and post-operative complications [14,15,16]. Furthermore, by using continuous data from wearables, changes in patient condition over time can be investigated instead of focusing on specific moments, which potentially allows to accurately capture patient recovery.

Therefore, we aim to create a model to predict daily post-operative recovery scores of patients in the hospital and after discharge at home using vital signs and activity metrics from wearable sensors. In this research, we utilized post-operative data from patients who underwent major abdominal cancer surgery. This patient population is characterized by a complex recovery trajectory, with high mortality rates and a high risk for complications [17]. Hence, better recovery assessment may be valuable for this patient group to improve patient management. Clinicians often have difficulties trusting complex models due to the absence of intuition and explanation of the outcomes [18]. Therefore, the model should not only obtain high prediction performance, but also maintain a sufficient level of interpretability to increase the chances of clinical adoption.

## 2. Materials and Methods

### 2.1. Study Details

Our study population consisted of a subset of the total population from the Transitional Care study 3 (TRICA). In the TRICA Study NCT03923127 (NL7602, PJ-013483 FLAGSHIP Transitional Care Study 3), data were collected using wearable sensors for post-operative monitoring of cancer surgery patients and bariatric patients. The study was approved by the ethical committee of the Maxima Medical Centre, Veldhoven, the Netherlands (W19.001). In total, 150 major abdominal cancer surgery patients were included in the TRICA study from April 2019 to August 2020. Several patients withdrew from the study, and the data of the remaining 125 cancer surgery patients were used to create and test a model for predicting daily recovery scores. More study details can be found in the publications of Jacobs et al. [19] and van der Stam et al. [20].

The data were recorded in the Catharina hospital in Eindhoven, the Netherlands, with two devices from Philips Electronics Nederland B.V. The two devices, a Healthdot sensor and an Elan sensor, were worn for approximately two weeks (Healthdot) or three weeks (Elan) post-surgery in the hospital and after discharge at home. The Healthdot was applied directly after surgery with an adhesive to the patient’s lower left rib on the mid-clavicular line. This sensor contains an accelerometer, and was developed for wireless remote monitoring of vital signs. The Healthdot measured the vital signs, heart rate (HR), and respiration rate (RR) and also recorded posture information and activity levels. The Healthdot measured continuously for 14 days, and the data were saved at eight-second (for HR) and one-second (for RR) intervals.

The Elan sensor was worn as a wristband. Each patient received two different Elan devices, which needed to be switched daily to charge the sensors. The Elan measured various activity features, including steps, walking speed, hours walking per day, activity counts (actcount), percentage active time, and the number of sedentary hours per day, derived from acceleration data, as well as HR and RR derived from photoplethysmography (PPG) signals. The activity counts metric detects the amount of motion per second using the acceleration data and transforms it into a unitless number. In addition, the percentage active feature was calculated with classifications for active and inactive signal parts, computed from the raw acceleration data. The Elan generated one additional activity feature, the active energy expenditure (AEE), which was based on acceleration data and the HR signal. The data were stored at one-second intervals.

### 2.2. Train-Test Split

We created a split in the patient group using two-thirds of the patients (*n*= 83) for developing and training a model and saving one-third of the patients (*n* = 42) for the testing phase [21]. The test data remained unseen until after feature extraction and model training. We aimed to obtain a balanced split in which we equally divided the variables age, gender, complications, LOS, and surgery type into two groups by splitting the patients 10,000 times. For each split, *p*-values were calculated with a chi-square test for categorical variables and a Kolmogorov–Smirnov test for numerical variables. The best-balanced division was found by multiplying the *p*-values from each variable and choosing the split with the highest product of *p*-values. Details of the split can be found in Table 1.

### 2.3. Reference Recovery Scores Engineering

In the TRICA dataset, no ground truth, the level of recovery over time for each patient, was available. Therefore, we had to engineer reference recovery profiles based on the literature, intuition, and patient events, such as discharge dates, complications, and readmission. These reference profiles were then used to assign a recovery score per day for all patients for training and validation purposes.

The reference recovery profiles follow an exponential trend, based on the assumption that considerable improvements in recovery are made in the first days, and after this, the recovery converges to a plateau [4,22]. The main factors influencing these profiles were the LOS and the complications, as shown in Figure 1. As can be seen in this figure, we created the recovery curves to increase from 1.0 to 2.0; 1.0 was defined as the starting point of recovery, while 2.0 was defined as fully recovered. On the day of hospital discharge, we assigned a score of 1.8, assuming that patients were not fully recovered on the day of discharge and continued recovery at home. Utilizing these assumptions, recovery profiles were obtained that depended on the LOS, as shown in Figure 1a.

When post-operative complications occurred either in the hospital or at home, we represented this with a drop in recovery score, as shown in Figure 1b. The size of this drop was based on the severity of the complication, by utilizing the Clavien–Dindo (CD) classification system [23]. The CD system contains grades from one to five, where a higher CD grade indicates a more severe complication. The drop for a CD grade of one or two was chosen as 0.02, for CD grade three as 0.04, and for CD grade four as 0.06. For a CD grade of four, we designed the recovery score to decrease two days before the complication, which was different compared to the less severe complications, for which we created a drop one day before the complication. This distinction was made because these severe complications were expected to be detectable earlier. If a complication occurred in the first few days post-surgery, the score dropped below 1.0.

### 2.4. Feature Extraction

We extracted features from both the Healthdot and the Elan sensor, after which we investigated the feature potential for predicting recovery by calculating the Pearson correlation with the reference recovery scores.

Daily features were calculated since we aimed to predict daily recovery scores. From the Healthdot sensor, the RR was utilized to calculate the mean and variance of the RR. The Healthdot-measured RR was preferred over Elan-measured RR because we considered the RR to be more reliable when measured at the ribcage than at the wrist. In addition, we utilized the activity level (actlevel) and the upright classification, defined in hours/day, of the Healthdot. Regarding the activity level, levels zero up to three indicate ‘in rest’, and levels four until 10+ represent activities with increasing intensities.

We preferred the HR signals from the Elan sensor over those of the Healthdot, as longer coverage was achieved (21 days) with the Elan compared to the Healthdot (14 days). Besides calculating the mean and the variance of the HR, we developed algorithms to calculate the resting heart rate (RHR) and the circadian rhythm (circ) of the heart rate. For the latter, Peak–Nadir features [24] were calculated or differences between the HR during the day and night were computed. Furthermore, heart rate variability (HRV) features were created in the time domain, such as the standard deviation of the normal-to-normal intervals (SDNN), the root mean square standard deviation of the normal-to-normal intervals (RMSSD), and the percentage of normal-to-normal intervals that differ more than 50 ms (pNN50). In addition, the inter-beat-interval data were transferred to the frequency domain with the Welch method [25], and the power was calculated in the very-low-frequency range (VLF), the low-frequency range (LF), and the high-frequency range (HF). From the LF and HF components, the LF/HF ratio was also calculated.

Features were also created that are a combination of activity and vital signs. Heart rate recovery was calculated, defined as the drop in HR after elevation by physical exercise up to at least 70% of the maximum patient HR (220-patient age [26]). Specifically, the drop in HR was calculated after one minute (HRR-1) and after two minutes (HRR-2), and the time it took for the HR to drop to ten beats per minute (bpm) (HRR-time) was also calculated. In addition, HR during activity features were analyzed; we calculated the HR during walking, upright, and sedentary periods and we calculated the HR during inactive and active periods, as well as the difference between those HRs. With a similar procedure, we calculated the HR during high and low Elan activity counts (threshold at 2000, the median activity counts for the patients) and high and low Healthdot activity levels (threshold at three, with ≤3 classified as ‘in rest’ and >3 as ’activity’).

In addition, delta features were created to add the history and the trend of the features to the model. These delta features were calculated by subtracting the feature value of the previous day from the value of the current day.

### 2.5. Machine Learning Model

For predicting daily recovery scores, we chose a machine learning approach since they frequently offer a high predictive performance, and machine learning is being increasingly used in the clinic [27]. As model input, we provided the daily features, and as target output, we provided the engineered daily reference recovery scores. The feature frame for each patient contained data for several days, and each day was treated as a new sample in the model. No information was supplied to the model about which days belong to the same patient or which samples belong to the first or the final days. The model was created in Python utilizing the xgboost library.

We explored multiple algorithms to train a regression model for predicting recovery scores, including Lasso regression, k-nearest neighbors, decision tree, random forest, support vector regression, and XGBoost, with different levels of interpretability. Details of the algorithms can be found in [28]. The algorithm performance was evaluated with the mean squared error (MSE), Spearman rank correlation coefficient (SRCC), and the error in recovery score at discharge. While all methods showed potential to model patient recovery, XGBoost consistently provided the highest performance (results can be found in [28]). Hence, the XGBoost algorithm was selected to train the final regression model, although it is more complex than, for example, Lasso regression and decision trees. To overcome the lower interpretability of XGBoost, we created a model with a low number of features. For this purpose, features with the lowest importance, as determined by the XGBoost algorithm, were removed until significant differences were found (Friedman test followed by a Nemeyi post hoc analysis) compared to the model with all features.

We obtained the optimal XGBoost hyperparameter values, such as the learning rate and the maximum tree depth, with five-fold cross-validation using the training set. The algorithm provided an embedded missing data method, in which samples were directed in the default direction if the feature needed for the split was missing. Many missing data points occurred, as the Elan sensor could be removed and was often not directly reapplied after charging or showering. We added early stopping to the algorithm to prevent overfitting and to enhance the model’s computational efficiency. Before providing the features to the model, we standardized them by scaling the data to a normal distribution with a mean of zero and a unit variance.

Figure 2 shows the machine learning pipeline of the model, in which all steps are visualized. In this figure, three stages can be detected: the data processing stage (1), the training stage (2), and the testing stage (3). In the first stage, a train–test split was created, the features were extracted, and delta features were calculated. The data of the independent test set were disregarded until the testing phase. The second phase is the training phase, with the training set and the reference recovery profiles as input. First, five-fold cross-validation was performed to select the best hyperparameter combination for the machine learning algorithms. Subsequently, we retrained the model with 10-fold cross-validation with the optimized hyperparameter values to investigate the model performance on the validation folds. The model with the highest-performing machine learning algorithm was then selected. Finally, features were removed to obtain the lowest number of features while maintaining the performance of the final model. This model served as the input of the testing phase, combined with the independent test set. On this test set, predictions were made for the recovery scores, after which we evaluated the model using the performance metrics.

## 3. Results

### 3.1. Feature Correlation with Reference Recovery Scores

The Pearson correlations between the features and the reference recovery scores are shown in Figure 3, with vital sign features in blue, activity metrics in orange, and combination features in green. Regarding the vital sign features, the mean and variance of HR and RR showed minor correlations of below 0.2, indicating they may not be strongly related to patient recovery. In contrast, the HRV features in the frequency domain and the circadian rhythm of the heart rate features showed stronger correlations, and are more likely to be important for predicting recovery. Strong correlations were also obtained for the activity features and some combination features, such as the HRR features.

Removing redundant features until significant differences in MSE and SRCC were found compared to the model with all features (*p* < 0.05) yielded a model with only six features: number of steps, upright hours, upright hours delta, circadian rhythm of the heart rate Day–Night, HRR-1, and HRR-1 delta.

### 3.2. Predictions on Independent Test Set

We tested the model with the final six features on the independent test set, the results of which are shown in Figure 4. Figure 4a shows the predicted profiles and their corresponding reference recovery curves for two patients. This figure demonstrates that the model could fit the curves of both a patient with a fast recovery (short LOS) in blue and a patient with a slower recovery (long LOS and complications) in red, although some fluctuation occurred compared to the smooth-engineered reference profiles. The predictions on the total test set are visualized in Figure 4b. The model achieved high performance, as a clear correlation (R = 0.85) existed between the predicted and reference recovery scores, and no clear bias was observed for under- or over-estimation of the scores. Furthermore, the performance metrics on the training and test sets in Figure 4c show that the model achieved comparable high performances on the training set as well as the test set, with MSEs of 0.014 and 0.021, SRCCs of 0.81 and 0.79, and discharge errors of 0.066 and 0.102, respectively. Hence, the model could be generalized to unseen patients and did not overfit the training data.

### 3.3. Usability in the Clinic

To identify patients with a slow or fast recovery, we compared the individual predicted profiles to the average profiles of the short LOS (≤8 days) and the long LOS (>8 days) patient groups. The value of eight was chosen since it equals the median LOS of the training group. For these average profiles, we also included the predictions on the training patients, such that 60 patients were included in the long LOS group and 65 patients in the short LOS group. Four examples of these comparisons are shown in Figure 5. Figure 5a visualizes a patient with high recovery scores on average, even in the first days after surgery. From these high scores, the patient can be identified as a member of the short LOS group, showing fast recovery. From day four onwards, predicted recovery scores higher than 1.8 were obtained, indicating that the patient was potentially discharge ready.

In contrast, Figure 5b,c shows patients with slower predicted recoveries, as the model provided low recovery scores for multiple days. Both patients obtained recovery scores comparable to the long LOS group. Low and occasionally decreasing recovery scores were also predicted before the occurrence of complications and readmission. Furthermore, according to the model, the patient who was readmitted (Figure 5c) was unready for discharge (recovery score < 1.8) on the day of discharge.

The model occasionally provided less useful predictions, as shown in Figure 5d. The predictions for this patient occurred often in the overlapping regions of the short and long LOS group. Furthermore, a complication occurred on day five, but the model did not obtain low or consistently decreasing recovery scores on the days before the event.

It is worth noting that the model predictions can be easily understood since the model only utilized six features. An example of a feature frame from the patient in Figure 5b is shown in Table 2. This patient obtained low recovery scores from day one to day three. As shown in the table for days one, two, and three, the patient was extremely inactive and their circadian rhythm parameter was low. Consequently, low recovery scores can be anticipated. Furthermore, from day seven onwards, the recovery scores decreased on average (aside from the increase on day 9), with a complication on day 11. The table shows that upright hours decreased over time and the circadian rhythm parameter was low. Abnormal values of the HRR-1 (lower than 12 bpm [26]), together with the decrease in activity and the low circadian rhythm, yielded low and decreasing recovery scores.

## 4. Discussion

With the developed model, we predicted recovery profiles that matched the reference profiles quite well for the patient test set, as shown by a high SRCC of 0.79, a low MSE of 0.021, and a low discharge error of 0.102. In addition, we showed that the model could be used to identify fast-recovering patients with high recovery scores equal to or higher than the short LOS group. By timely identification of fast recovery, our model may help to schedule appropriate discharge dates. Similarly, we were able to identify slow-recovering patients with low recovery scores equal to or lower than the long LOS group. These patients are likely to have a late discharge and may require additional care. For some patients, the model showed low recovery scores the days before the post-operative complication, so identifying such events might be possible, even though this was not the focus of our research. After identifying a slow-recovering patient, clinicians can investigate why the patient obtains low recovery scores and, if it is needed, can schedule more frequent check-ups.

Utilizing the discharge readiness threshold of the model, readmission might be prevented, as, for example, the model predicted that the patient with readmission in Figure 5c was unready for discharge on the day of discharge. Besides identifying patients in need of extra care who might develop complications and predicting appropriate discharge dates, the model may also be utilized to detect potential deterioration at home. Furthermore, the model might facilitate planning of resources, surgeries, rehabilitation, and available beds.

The model was less informative for some patients; for example, patients who obtained recovery scores in the overlapping region of the short and long LOS group were more challenging to classify as slow or fast recovering. Furthermore, before a complication, the model occasionally predicted no low or decreasing recovery scores, limiting the model’s ability to consistently predict such events. However, we can monitor the general trend in patient recovery utilizing the model.

This research shows the potential for predicting discharge readiness, detecting complications, preventing readmissions, and planning resources using continuous recovery scores from data of wearable sensors. The results indicate that physicians could utilize this method for the support of recovery-based decisions in the hospital and at home.

From the data of the wearable sensors, we extracted different features and we determined the features’ importance to post-operative recovery. Activity features, circadian rhythm of the HR, and HRR showed the highest importance to predict post-operative recovery, whereas the HR and RR features were less important. This emphasizes the importance of implementing additional metrics in the clinic than the current metrics utilized for recovery-based decisions. Moreover, by incorporating only a limited number of features in the final model, it is not difficult to interpret the results. The final features were a combination of activity, HR-based metrics, and delta features. Consequently, also the trends in certain features improved the predictive performance.

To overcome the limitations of current methods to monitor post-operative patient recovery, such as the high level of subjectivity [3], the focus on predicting only one event [8,9], and infrequent measurement, we created a model to predict daily recovery scores from continuous data. This model provides a novel automated technique for monitoring post-operative recovery with the use of data from wearable sensors. The proposed method is neither labor intensive nor time consuming, and it provides continuous predictions. The model’s results, therefore, go beyond the state-of-the-art solutions. For example, for existing EWSs, parameters are measured infrequently, whereas with our approach, patient data are continuously measured, thereby potentially making its predictions robust.

### 4.1. Outlook

We envision predictive modeling will be be used in combination with existing methods for recovery assessment. It can be utilized as an interpretable, automatic, and objective input to support recovery-based decisions. For example, if low recovery scores are obtained, more spot checks can be scheduled. Analogously, if high recovery scores arise, fewer spot checks can be scheduled. In addition, clinicians can provide the patients with the lowest recovery scores with additional care.

Alarming can be implemented if the patients obtain low or high recovery scores or if recovery scores decrease for several days. This alarm increases the recovery score usability in the clinic, as daily investigation of all patient profiles is time consuming. However, future research is needed to investigate how to set alarm boundaries for low and high recovery scores with a certain confidence level. Furthermore, it would be interesting to investigate if the model can be trained to specifically predict the occurrence of discharge and complications.

### 4.2. Comparison to the Literature

Machine learning and other modeling techniques to support clinical decision making are receiving substantial attention in current research. Many investigations are focused on predicting events and outcomes, such as risk of complications, discharge, length of stay, or mortality [8,9,29,30,31,32]. A limitation of some of the previous studies is that they only include demographic variables and/or parameters that are measured a limited number of times, leading to an inability to adjust predictions based on new information [11,29,32,33].

For example, in a review by Campagnini et al. on the use of machine learning methods to predict patient recovery in stroke patients, the research goal in general was to predict patient-specific functional performance at a single future point in time based on data such as patient demographics, medical history, and assessment results rather than continuous data [34]. Nevertheless, Campagnini et al. concluded that these predictive models can be a very promising support tool for clinicians, as they can provide accurate recovery estimates at a low cost. A use case where predictions are updated more regularly throughout the patient hospital stay is EWSs. Muralitharan et al. stated that machine-learning-based EWSs, which have the ability to include continuous vital sign data, are more effective in predicting physiological patient deterioration compared to traditional aggregate-based risk stratification tools [35]. However, false alarming of such models is mentioned as an important attention point to improve clinical adoption [35].

An interesting example of the use of modeling for discharge predictions is the recent work of Cai et al. [36]. They used a Bayesian network model to estimate the probability of a hospitalized patient being at home, in the hospital, or dead on each of the next 7 days, utilizing patient-specific administrative and laboratory data and updating predictions when new data became available. This enables real-time forecasts for patient outcomes, thereby providing richer information than traditional point predictions of length of stay, death, or readmission, which may better support decision making [36].

Overall, it is clear that predictive modeling holds promise to support clinical decision making in a variety of settings. A distinction between our work and the discussed research is that our focus was on predicting the patient’s current recovery status rather than future events. By combining continuous vital sign and activity data into a single objective and interpretable daily recovery score, clinicians can easily compare patient scores to those of relevant patient groups and see patient progress compared to previous days. This allows them to take into account a larger amount of data when assessing the patient’s health status, without needing to look into daily graphs of vital signs or activity data. Based on comparisons to other research, it could be interesting to investigate inclusion of additional data such as lab test results, blood pressure, and demographics to improve the prediction accuracy, although including more parameters might also decrease model interpretability. Furthermore, while models that use lab results, spot checks, and clinical assessments as inputs are often limited to the hospital stay, our model leveraging data from wearables provides continuation of recovery scores after discharge, allowing deterioration detection at home as well.

### 4.3. Future Work

We utilized the short and long LOS groups as the reference average patient population; however, more than two groups can be used to obtain more distinct patient subpopulations. Furthermore, we used the model to generate daily recovery scores, whereas predictions in a shorter period could also be of interest, for example, if aimed at detecting complications. Furthermore, it is worthwhile to note that although the XGBoost algorithm showed the highest performance, predicting recovery scores is not limited by the use of this algorithm. Other algorithms might obtain similar predictions, even non-machine learning algorithms, such as rule-based methods.

Future research is required to investigate the generalizability to patient groups with different surgeries. If such generalization is possible, the recovery score prediction model can be used in multiple wards in the hospital. Furthermore, the proposed method of predicting continuous recovery scores using data from wearable sensors could potentially be valuable for other use cases, such as recovery after cardiac infarction or a disease, even if the model itself or relevant parameters may be different.

### 4.4. Limitations

The findings of this research may be somewhat limited by the created reference recovery profiles. We based the reference profiles on discharge dates, complications, and assumptions. The discharge date was a significant contributing factor, but this event is also influenced by factors other than patient health status. For example, the availability of a nursing home placement could impact the discharge of certain patients. Moreover, we designed the reference recovery profiles to be smooth, whereas real-time recovery is likely to fluctuate over time. For future studies, it could be worthwhile to define a golden standard with fewer uncertainties and assumptions. For example, doctors’ judgments, in the form of a recovery score, could be obtained daily to judge the patient’s health during their stay in the hospital.

## 5. Conclusions

In this study, we created a model that uses activity and vital sign data from wearables to determine day-to-day post-operative recovery scores in the hospital and at home which can be applied to support recovery-based decisions. Such models can be combined with current solutions in the hospital as an objective measure to support a physician’s judgment. The model achieved a high predictive performance and at the same time provided predictions that were interpretable. By comparing patient-specific predicted profiles of slow- and fast-recovering patients of a similar population, patients could be identified with fast and slow recovery. Moreover, our results indicated that in addition to traditionally measured parameters such as heart rate and respiration rate, other metrics, such as activity parameters, circadian rhythm of the heart rate, and heart rate recovery, could be highly relevant for monitoring patients. The model may be limited by the definition of the reference recovery curves, which may be solved by implementing a different recovery golden standard in future research. Nevertheless, the proposed method has shown the potential to measure post-operative recovery in an automated and continuous way. With this method, we might be able to predict discharge, detect complications, prevent readmissions, and plan rehabilitation.

## Figures and Tables

**Figure 1 sensors-23-04455-f001:**
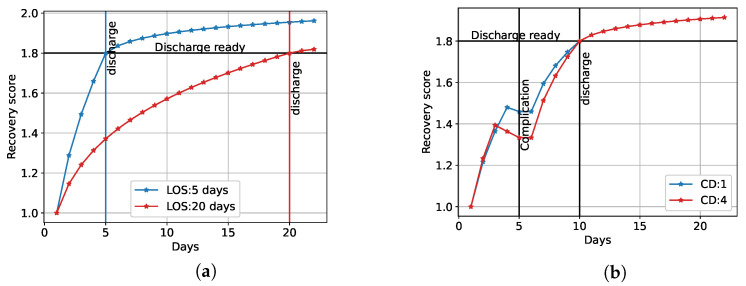
(**a**) Engineered reference recovery profiles for two patients without complications and a different LOS. (**b**) Engineered reference recovery profiles for two patients with complications on day 5 and discharge on day 10. The complications differ in severity (CD scores).

**Figure 2 sensors-23-04455-f002:**
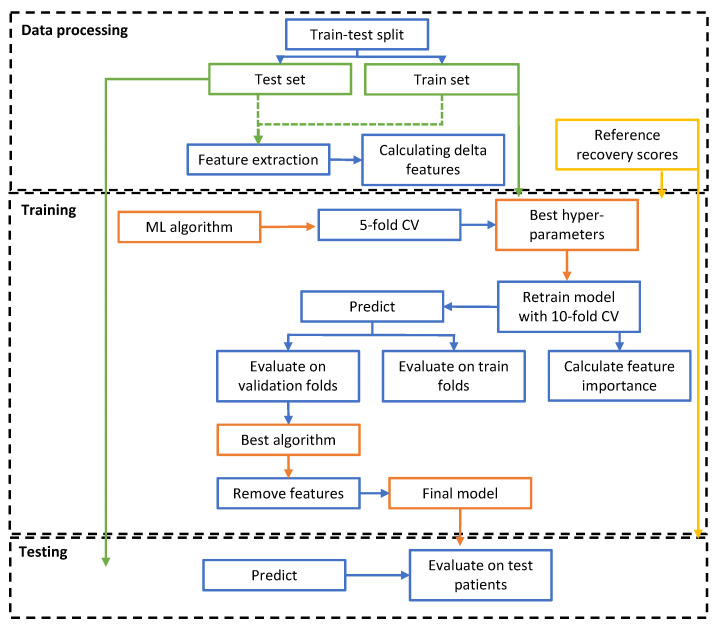
Machine learning pipeline with three stages: data processing, training, and testing. Datasets are displayed in green, the computational methods in blue, the reference standard in yellow, and the model elements in orange.

**Figure 3 sensors-23-04455-f003:**
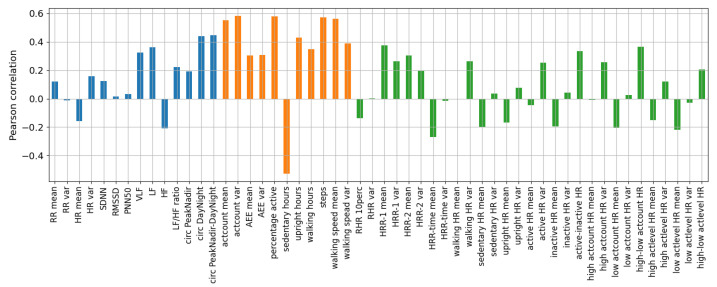
Pearson correlation between the reference recovery scores and the created features per day for the training set. Vital signs are displayed in blue, activity metrics in orange, and combination features in green.

**Figure 4 sensors-23-04455-f004:**
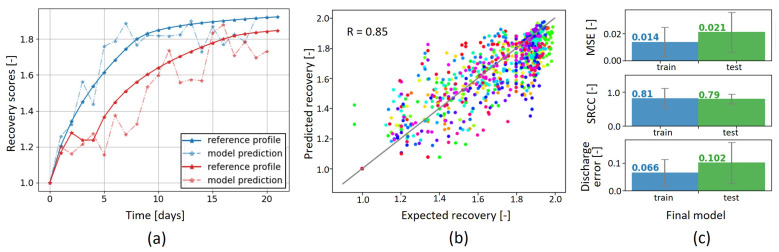
Predictive performance of the regression model. (**a**) Individual predicted profiles for two patients. (**b**) Scatterplot with all predictions on the test set; each daily prediction is a dot and each color represents a patient. (**c**) Performance metrics on the training set (blue) and test set (green).

**Figure 5 sensors-23-04455-f005:**
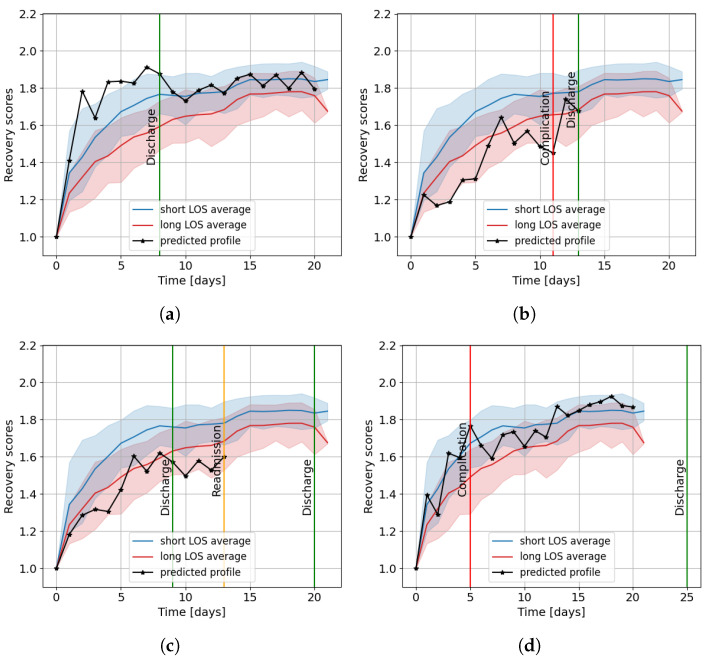
The average profiles of the long LOS and short LOS groups of the entire dataset compared to the predicted individual profiles of a patient (**a**) with high recovery scores, (**b**) with low recovery scores and a complication, (**c**) with low recovery scores and a readmission, and (**d**) with average recovery scores and a complication. The error bands of the average profiles are shown as the 10th and 90th percentile, and the recorded events are visualized with vertical lines.

**Table 1 sensors-23-04455-t001:** Division of the patients into a training and a test set. The categorical variables are displayed by the number of patients and the percentage and the continuous variables are displayed by the mean. The different surgery types are esophageal resections (ER), hyperthermic intraperitoneal chemotherapy (HIPEC) surgeries, pylorus-preserving pancreatoduodenectomy (PPPD) or Whipple surgeries, low anterior resection (LAR) with intraoperative radiotherapy (IORT), and LAR without IORT.

Variable	Training	Test
Total patients	83 (66.4%)	42 (33.6%)
Female	38 (66.7%)	19 (33.3%)
Male	45 (66.2%)	23 (33.8%)
Patient with no complications	55 (66.3%)	28 (33.7%)
Patient with one complication	22 (66.7%)	11 (33.3%)
Patient with two complications	4 (66.7%)	2 (33.3%)
Patient with three complications	2 (66.7%)	1 (33.3%)
Surgery: ER	14 (63.6%)	8 (36.4%)
Surgery: HIPEC	13 (56.5%)	10 (43.5%)
Surgery: PPPD/Whipple	13 (76.5%)	4 (23.5%)
Surgery: LAR with IORT	13 (68.4%)	6 (31.6%)
Surgery: LAR without IORT	3 (75%)	1 (25%)
Surgery: Other	27 (67.5%)	13 (32.5%)
Age (mean)	62.2	62.1
LOS (mean)	11.3	10.3

**Table 2 sensors-23-04455-t002:** Feature values of the patient in Figure 5b. No HRR-1 is detected in the first days, as the patient is inactive.

Days	Number of Steps per Day (-)	Upright (Hours/Day)	Upright Delta	Average HRR-1 (bpm)	HRR-1 Delta	Circ. Rhythm Day–Night (bpm)
Day 1	20	0	0	inactive	-	2.08
Day 2	85	0	0	inactive	-	6.76
Day 3	141	0	0	inactive	-	6.3
...						
Day 7	1158	0.85	−0.09	12	1.7	9.23
Day 8	1084	0.77	−0.08	14.3	1.3	−2.3
Day 9	1438	0.77	0	8.2	−6.1	2.00
Day 10	809	0.34	−0.43	13.6	5.4	2.70
Day 11	1302	0.51	0.17	7.6	−6	0.62

## Data Availability

Restrictions apply to the availability of these data. Data were obtained at Philips Research and the Catharina Hospital and are available with the permission of Philips Research and the Catharina Hospital.

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
