# Peer review of "Machine Learning for Postoperative Continuous Recovery Scores of Oncology Patients in Perioperative Care with Data from Wearables"

_sensors, 2023, doi:10.3390/s23094455_

Round 1

Reviewer 1 Report

The paper overall is good, but it needs minor improvements 

1-Please add a comparison with other work section

2- please add limitation and future work sections. 

3- please add table in related work section to compare between literature 

Author Response

English: changes are made to improve the language

1-Please add a comparison with other work section

Added other work section in the discussion, called ''comparison with literature''

2- please add limitation and future work sections. 

Added limitation and future work section to the discussion

3- please add table in related work section to compare between literature

In consultation with the other authors, we decided to not add this table. Instead, we added a broad comparison with literature written in text. This is also more in line with the feedback of the other reviewers.

Thank you for your feedback.  

Reviewer 2 Report

Comparision of explored machine learning algorithms can be added

More elaboration for best suited XGBoost algorithm make more it readable

Detail of the machine learning algorithms can be added in terms of mathematical model 

Discussion can be divided into sub -sections for better readability

Work can be compared  with earlier research work

Author Response

English: english is reviewed again and minor changes are made to make the language better

1- Comparision of explored machine learning algorithms can be added

We cited my master's thesis which contains the comparison of the machine learning algorithms. We did it this way because the goal of the paper is not to show which algorithm is best, but to show that we might be able to predict recovery with machine learning. The machine learning algorithm can be interchangeable and similar results can be obtained.

2- More elaboration for best suited XGBoost algorithm make more it readable

More explanation of the chosen XGBoost algorithm is added in the methods section 'machine learning model'. 

3- Detail of the machine learning algorithms can be added in terms of mathematical model 

We cited my master's thesis which contains the mathematical details of all the machine learning algorithms.

4- Discussion can be divided into sub -sections for better readability

We divided the discussion into several sections: general, outlook, comparison to literature, future work, and limitations 

5- Work can be compared  with earlier research work

We added a section 'comparison to literature' in the discussion which reviews some previous studies in detail, and compares them with our study.

Thanks for your feedback. 

Reviewer 3 Report

1. Line 110-111: Kindly provide the citation and/or justification for the proportion used for the testing-training split.

2. Before the inputs are being fed to the machine learning models, are they subjected to any normalization or mathematical transformation/scaling?

3. Figure 2 is a very good illustrative diagram that summarizes the methodology adopted.

4. It would be interesting to extend the discussion section by including a paragraph detailing the comparison of the proposed methods and other comparable existing methods.

5. Kindly include more recent citations (2020 – 2023).

Author Response

English: We reviewed the paper again and minor changes are made to improve english language in the paper.

  1. Line 110-111: Kindly provide the citation and/or justification for the proportion used for the testing-training split.
    Citation added to adjust for the proportion.
  2. Before the inputs are being fed to the machine learning models, are they subjected to any normalization or mathematical transformation/scaling?
    Sentence added in materials and methods section:  Before providing the features to the model, we standardized them by scaling the data to a normal distribution with a mean of zero and a unit variance.
  3. Figure 2 is a very good illustrative diagram that summarizes the methodology adopted.
    Thank you for this positive feedback
  4. It would be interesting to extend the discussion section by including a paragraph detailing the comparison of the proposed methods and other comparable existing methods.
    Section added in the discussion 'comparison with literature' to explain the main difference between the current study and other studies. Also, the last paragraph before the section 'outlook' compares the method with current methods in healthcare.
  5. Kindly include more recent citations (2020 – 2023).
    We added 5 papers between the years 2020 and 2023

Thank you for your feedback.